# Adverse Effect of the Duration of Antibiotic Use Prior to Immune Checkpoint Inhibitors on the Overall Survival of Patients with Recurrent Gynecologic Malignancies

**DOI:** 10.3390/cancers15245745

**Published:** 2023-12-07

**Authors:** Hye-Ji Jung, Jong-Ho Park, Jina Oh, Sae-Mi Lee, Il-Yeo Jang, Jung-Yong Hong, Yoo-Young Lee, Hyun Jin Choi

**Affiliations:** 1Gynecologic Cancer Center, Department of Obstetrics and Gynecology, Samsung Medical Center, Sungkyunkwan University School of Medicine, Seoul 06351, Republic of Korea; hj13.jung@samsung.com (H.-J.J.); sammie14.lee@samsung.com (S.-M.L.); ilyeo.jang@samsung.com (I.-Y.J.); 2Chung-Ang University College of Medicine, Seoul 06974, Republic of Korea; dvm7799@gmail.com; 3Department of Obstetrics and Gynecology, Chung-Ang University Hospital, Chung-Ang University College of Medicine, Seoul 06974, Republic of Korea; oja881011@cauhs.or.kr; 4Division of Hematology-Oncology, Department of Medicine, Samsung Medical Center, Sungkyunkwan University School of Medicine, Seoul 06351, Republic of Korea; jungyong.hong@samsung.com; 5Department of Obstetrics and Gynecology, Chung-Ang University Gwangmyeong Hospital, Chung-Ang University College of Medicine, Gwangmyeong-si 14353, Republic of Korea

**Keywords:** antibiotics, body mass index, immune checkpoint inhibitors, gynecologic malignancies, overall survival

## Abstract

**Simple Summary:**

Recent studies have shown a negative association between prior antibiotic use and response rate to immune checkpoint inhibitors (ICIs) in solid tumors, including gynecologic cancers. In this study, use of antibiotics for >14 days prior to ICI treatment was associated with reduced survival. Restricted use and adequate duration and spectrum of antibiotics should be considered when treating patients with recurrent gynecologic cancer. The various response rates to ICI could be attributed to gut dysbiosis. A study of gut microbiota may address the ICI response in the future.

**Abstract:**

Purpose: Antibiotic use preceding immune checkpoint inhibitor (ICI) treatment has been associated with a decreased efficacy of ICI in solid tumors. In this study, we evaluated the effect of antibiotic use before ICI therapy on oncological outcomes. Methods: We examined patients with recurrent gynecologic malignancies at two academic institutions. The clinical data, including antibiotic use within 60 days of ICI initiation, type of antibiotics, reasons for antibiotic use, body mass index, tumor site, chemotherapy-free interval, prior history of radiotherapy, disease control rate (DCR), and overall survival (OS), were assessed. Results: Of 215 patients, 22.9% (*n* = 47) received antibiotics before ICI treatment. The most common cancer was ovarian (52.1%, *n* = 112), followed by cervical (24.7%, *n* = 53) and endometrial (16.7%, *n* = 36). When we divided the cohort based on antibiotic use before ICIs, there were no significant differences in the DCR and baseline characteristics between the two groups. On multivariate analyses, the variables associated with poor OS were previous use of antibiotics for a cumulative duration of >14 days (HR 2.286, 95% CI 1.210–4.318; *p* = 0.011); Eastern Cooperative Oncology Group 2 or 3 (HR 4.677, 95% CI 2.497–8.762; *p* < 0.001); and chemotherapy-free interval of <6 months (HR 2.007, 95% CI 1.055–3.819; *p* = 0.034). Conclusion: Prior use of antibiotics for a cumulative duration of >14 days was associated with reduced survival in recurrent gynecologic malignancies.

## 1. Introduction

The efficacy of immune checkpoint inhibitors (ICIs) has been reported, with particular features, such an extended duration of response and survival in patients with some solid cancers [1,2,3]. In the KEYNOTE-158 study on recurrent gynecologic cancer, cases of cervical and ovarian cancer were demonstrated to have modest response rates to pembrolizumab, with objective response rates (ORRs) of 14.3% and 19.0%, respectively [4,5]. On the other hand, encouraging results have been reported for endometrial cancer. The reported ORR was 48% for microsatellite instability high (MSI-H) endometrial cancer tumors that were recurrent or in the advanced stage [6]. 

In addition to tumor site, other factors affect the response to ICIs [1,2,3,4,5]. Even for the same tumor site, a subgroup of patients can show long-term survival of up to 10 years, whereas others may not benefit because of poor response or life-threatening immune-related adverse events, such as pneumonitis, myocarditis, or hepatitis [2,7]. Therefore, it is crucial to identify the optimal candidates who can benefit from ICIs. To date, high programmed death ligand 1 (PD-L1) expression, mismatch repair deficiency (MMRd), MSI-H, and tumor mutation burden have been considered the predictive biomarkers for improved patient outcomes [6,8,9]. However, the range of response rates remains wide among patients with the same predictive biomarkers; this highlights the need to identify additional biomarkers to predict response to ICIs. 

Recently, studies have shown a negative association between prior antibiotic use and response rate to ICIs in solid tumors, including gynecologic cancers [10,11]. One possible explanation for the decreased response to ICI is the use of broad-spectrum antibiotics, which may affect the diversity and composition of the gut microbiota and result in dysbiosis. Recovery of antibiotic-induced dysbiosis of the intestinal microflora can take 6 weeks to 6 months [12,13].

This study aimed to evaluate the effect of antibiotics use before ICI therapy in recurrent gynecologic cancer. The effect of antibiotics on ICI response was assessed according to the disease control rate (DCR) and overall survival (OS) in women with recurrent gynecologic cancers.

## 2. Materials and Methods

### 2.1. Patient Selection

After institutional review board approval (No. 2022-01-142-001 and No. 2206-023-19425), patients who were treated with ICIs for recurrent or metastatic gynecologic malignancies in the ovary, cervix, uterus, vagina, or vulva at two academic centers (Samsung Medical Center and Chung-Ang University Hospital) were identified. All the patients received the anti-programmed cell death protein 1 monoclonal antibody (antiPD-1 mAb) pembrolizumab or nivolumab alone. Since the time of ICI availability in Korea, the indications for pembrolizumab have included recurrent gynecologic malignancies (cervix, ovary, and endometrium) with PD-L1 positivity or MMRd/MSI-H or recurrent cervical cancer with squamous cell histology. Single-agent treatment with nivolumab was indicated for cases of recurrent squamous cell carcinoma of the cervix, vagina, or vulva or cases of platinum-resistant ovarian cancer with three or more prior lines of chemotherapy, regardless of the status of PD-L1 expression or MMRd/MSI-H. When either of the two antiPD-1 mAbs was indicated, the choice was based on physician preferences. Patients who were treated with a combination therapy of ICIs and any targeted agents or chemotherapy were excluded from the study.

### 2.2. Data Collection

The patient electronic medical records were reviewed to investigate the oral or intravenous antibiotics used within 60 days before the initiation of antiPD-1 monotherapy. The clinical data, including the type, duration, and reasons for antibiotic use; body mass index (BMI); tumor site; chemotherapy-free interval; prior history of radiotherapy; MMR/MSI-H status, Eastern Cooperative Oncology Group (ECOG) performance status; PD-L1 expression; and best objective response at 8–10 weeks and OS, were assessed. Response to ICI therapy was classified according to the iRECIST criteria [14]. 

### 2.3. Assessment of PD-L1 Expression and MMR/MSI Status

The tumor MMR status was assessed by IHC staining of four MMR enzymes (MLH1, MSH2, PMS2, and MSH6). Tumors that displayed loss of MMR expression in at least one of these four markers were categorized as MMRd. The MSI status was determined using a polymerase chain reaction (PCR)-based MSI analysis of DNA that was extracted from both normal and tumor tissues. This analysis was carried out using one of five mononucleotides loci (BAT25, BAT26, NR21, NR24, and Mono27) alone or in combination with dinucleotide loci and dinucleotide (BAT25, BAT26, D17-S250, D2S123, and D5S346), following the established protocol of the institution. Specimens were classified as MSI-H when at least two allelic loci among the five microsatellite markers examined shifted in size. The tumors were labeled as MMRd/MSI-H if they exhibited either MMRd or MSI-H. 

To assess the tumor PD-L1 expression, we used the PD-L1 IHC 22C3 antibody (Agilent Technologies, Inc., Santa Clara, CA, USA). This assessment yielded the tumor proportion score (TPS), which was defined as the percentage of viable tumor cells that expressed PD-L1. Alternatively, using the PD-L1 IHC22C3 pharmDx assay (Agilent Technologies, Inc., Carpiteria, CA, USA), the combined positive score (CPS) was calculated by dividing the number of PD-L1-positive cells, including tumor cells, lymphocytes, and macrophages, by the total number of viable tumor cells, multiplied by 100. PD-L1 positivity was defined as a TPS of ≥1% or a CPS of >1% [15].

### 2.4. Assessment of ICI Response

Before the start of treatment, baseline tumor assessment was performed. Response was evaluated by abdominopelvic and/or chest computed tomography scans every two or three cycles of ICI treatment. Tumor response was assessed by a gynecologic oncologist at each institution, according to the Response Evaluation Criteria in Solid Tumors (RECIST) ver. 1.1. The primary endpoint was DCR, which was determined by the proportion of patients with complete response (CR), partial response (PR), or SD based on the RECIST ver.1.1. The secondary endpoints included the progression-free survival (PFS), which was defined as the time from the start of treatment to tumor progression or death, whichever occurred first, and the OS, which was defined as the time from the start of treatment to death from any cause.

### 2.5. Statistical Analysis

The normality of the data was evaluated using the Shapiro–Wilk test. Normally distributed data were reported as mean ± standard deviation, whereas nonnormally distributed data were reported as median (interquartile range). The frequencies of the categorical variables were compared using the chi-square test or Fisher’s exact test. Three-group comparison was performed with the analysis of variance test. Quantitative variables were compared using one-way analysis of variance as a parametric test or the Kruskal–Wallis test as a nonparametric test. Survival curves were calculated using the log-rank test, according to the Kaplan–Meier method. The Cox proportional hazards model was used for multivariate analysis to assess the independent prognostic factors. Statistical significance was set at *p* < 0.05. Statistical analysis was performed using IBM SPSS Statistics for Windows, Version 25.0. (IBM Corp, Armonk, NY, USA).

## 3. Results

Of the 215 eligible patients, 22.9% (*n* = 47) received antibiotics before ICI treatment (Table 1). The mean BMI was 22.2 ± 3.9 kg/m^2^, and 17.2% (*n* = 37) patients had low BMI (<18.5 kg/m^2^). The most common cancer was ovarian (52.1%, *n* = 112), followed by cervical (24.7%, *n* = 53) and endometrial; (16.7%, *n* = 36); 66% of the patients had ECOG 2 or 3. The median number of previous lines of chemotherapy was 3 (range, 1–11), and the median chemotherapy-free interval was 2 months (range, 0–54 months); 51.2% had prior radiotherapy. The ICI received was pembrolizumab in 85.1% or nivolumab in 14.9%. The age, BMI, ECOG, PD-L1 expression, dMMR/MSI-H, tumor sites, number of previous lines of chemotherapy, chemotherapy-free interval, prior radiotherapy, and type of ICI were not significantly different between the group that had used antibiotics before ICIs and the group that did not use antibiotics. 

The DCR at 8–10 weeks after ICI treatment (Figure 1) was 33.3% in the patients without prior antibiotics and 23.4% in those with previous use of antibiotics (*p* = 0.130); among the latter, the DCR did not significantly change, regardless of the duration of antibiotic use. The percentage of patients with progressive disease tended to be higher in those with low BMI than in those with high BMI (>24.9 kg/m^2^) (78.4% vs. 56.6%, *p* = 0.065; Table 2). The OS was worst in patients who had used antibiotics for ≥14 days and was worse in patients with low BMI than in those with normal or high BMI (Figure 2).

In addition to prior use of antibiotics and BMI, the MMR/MSI status (*p* = 0.007), chemotherapy-free interval (*p* = 0.05), and ECOG status (*p* < 0.001) were significantly associated with OS (Appendix A). Compared with the other patients, those who had endometrial cancer showed a tendency for the best survival rates (*p* = 0.065, Appendix A). Furthermore, prior history of radiotherapy, type of ICI, PD-L1 expression, and previous lines of chemotherapy were not significantly associated with OS (Appendix A). On multivariate analysis for OS, prior antibiotic use for ≥14 days, a chemotherapy-free interval of <6 months, and poor ECOG were the independent factors for worse OS (Table 3).

The most common reasons for antibiotic use were urinary tract infections, procedure-related indications, and peritonitis (Table 4). However, the durations of antibiotic use varied widely in this study population. The most common reason for antibiotic use for ≥14 days was procedure-related indications, followed by sepsis.

## 4. Discussion

The present study demonstrated that survival was associated with prior antibiotic use for ≥14 days before ICI therapy. In addition, a chemotherapy-free interval of <6 months and poor ECOG status were independent prognostic factors associated with decreased survival after ICI therapy. The most significant differences in the response patterns between ICI and cytotoxic chemotherapy are delayed response, durable response, and OS gain, regardless of the PFS or ORR [16,17,18]. This pattern was also observed in the present study. The BMI and duration of antibiotic use prior to ICI therapy did not significantly affect the DCR, but they significantly affected the OS. Moreover, the duration of antibiotic use was an independent prognostic factor, in terms of OS, after ICI therapy. 

Although MMR/MSI status has previously been identified as the most potent prognostic factor [19], it was not an independent prognostic factor in our study. Both MMR/MSI status, in 98 patients (45.6%), and endometrial cancer were significant prognostic factors for survival after ICI in the univariate analysis but not in the multivariate analysis. Because of the relatively small sample size of patients with known MMR/MSI status, our study did not have the power to detect any survival differences. Testing for MMR/MSI status is usually performed upon endometrial cancer diagnosis but is not routine for cervical and ovarian cancers. Therefore, the dMMR/MSI-high group was mainly derived from the patients with endometrial cancer. In cervical cancer, MMR/MSI status is usually not tested because the use of ICIs is approved regardless of MSI/MMR or PD-L1status. In most ovarian cancers, only PD-L1 expression is tested, because ICIs are approved in patients with PD-L1-positive tumors. 

The decreased survival of patients who received antibiotics for ≥14 days may be attributed to dysbiosis of the gut microbiota, which is an important modulator of ICI activity. In fact, patients who were treated with ICIs were reported to have significant differences in response rates, according to their gut microbial diversity and composition [20,21]. Moreover, the most frequently used antibiotics in this study population were cephalosporins, which are ineffective against Bacteroidetes, which had been reported to be abundant in non-responders to ICIs [22]. 

Our study evaluated the effect of antibiotic use within 60 days before the initiation of antiPD-1 monotherapy. A previous study reported a negative effect of prior antibiotic use within 30 days on ICI therapy [10]. Antibiotic-induced dysbiosis takes 6 weeks to 6 months to recover [12,13]. Therefore, we decided to investigate extended periods of previous antibiotic use. In addition, we described the cause of the antibiotics use equal or longer than 2weeks. In 8 of 48 patients with prior use of antibiotics, the indication was preprocedural prophylaxis; of these, 6 were administered antibiotics for ≥14 days. This finding implied the need to consider the adequacy of prophylactic antibiotic use in this setting.

Our study had several limitations that must be considered when interpreting the findings. First, because of the relatively small sample size, we could not perform subgroup analyses of each clinical factor such as disease site or type of infection thoroughly as in a prospective clinical trial. In particular, the number of endometrial cancer cases that were treated with antibiotics was too small to analyze the effect of ICI therapy on survival. When we examined the primary tumor site in detail, the prior antibiotic use group comprised only 4 (8.6%) patients with endometrial cancer but 80.8% patients with cervical and ovarian cancer. Therefore, the effect of prior antibiotic use on patients with endometrial cancer could not be concluded in this study. A larger study is needed to confirm the effects of antibiotic use prior to ICI therapy for subgroup analysis. 

Second, the lack of available data from the retrospective study design likely contributed to the study population’s heterogeneous clinical factors and biomarkers. PD-L1 and MSI/MMR status was not fully evaluated. Therefore, this study cannot evaluate the clinical benefit of PD-L1 expression and dMMR/MSI-H. The effect of the previous anti-cancer therapies on ICI therapy with previous antibiotic use was not analyzed because of their heterogeneity. Although stool sampling is essential for analyzing the gut microbiota, the sampling was not performed because of the retrospective study design. Investigators of this study are conducting ongoing research on the microbial composition of responders and non-responders to ICIs, according to antibiotic treatment. 

Third, the cutoff of 14 days for the duration of antibiotic use was arbitrary. The type and appropriate duration of antibiotic use are determined by the type and severity of the infection. In cases of prolonged antibiotic use, the duration was dependent on the predisposing factors to infection (e.g., weakened immunity, indwelling catheter or drain, or advanced-stage disease). These clinical situations may have confounded the shorter OS with longer antibiotic use.

Despite its limitations, this study provided important information on the effect of prior antibiotic use on the OS of patients who were treated with ICIs. This study added evidence that prior long-term antibiotic use, in addition to the known biomarkers, may be a predictive marker for adverse survival in patients treated with ICIs. A phase I study demonstrated that antiPD-L1-refractory melanoma responded to ICI after fecal microbiota transplantation (FMT) [23,24]. In antibiotic-induced gut microbiome dysbiosis, FMT may be an option to improve the response to ICI. In addition, probiotics, prebiotics, diet, and lifestyle may modulate the gut microbiota [25,26,27,28,29,30,31]. 

## 5. Conclusions

Gynecologic oncologists should consider the adverse effects of antibiotics on the prognosis of patients treated with ICIs. Restricted use, as well as adequate duration and spectrum of antibiotic use, should be considered. In addition, if modulation of the gut microbiota is proven to improve the effect of ICIs in recurrent gynecologic cancers, FMT may be a treatment option for patients with dysbiosis before ICI therapy. 

## Figures and Tables

**Figure 1 cancers-15-05745-f001:**
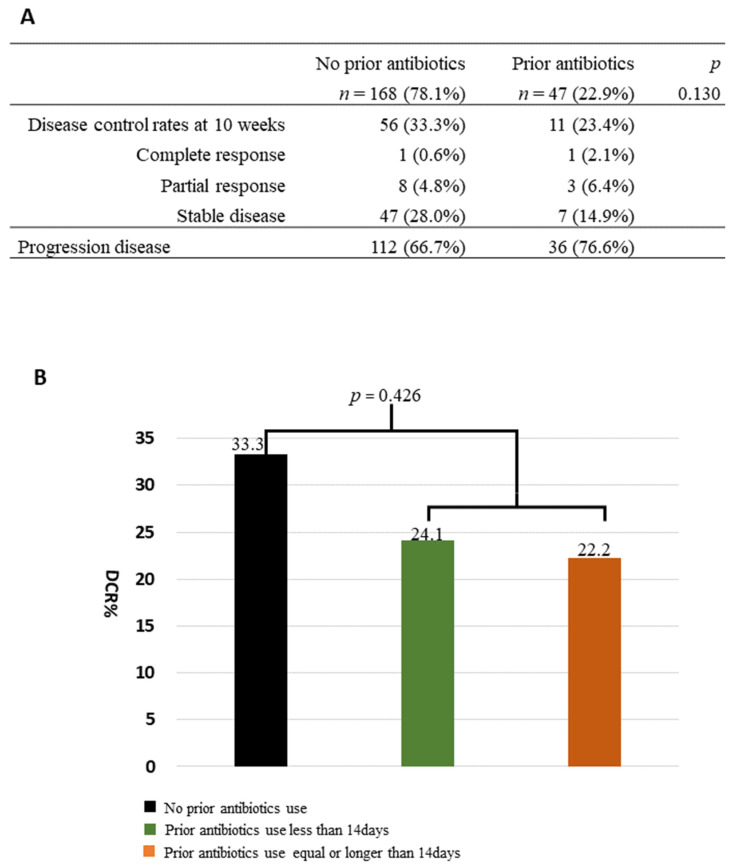
Disease control rates at 8–10 weeks. DCR of recurrent gynecologic cancers after ICI treatment (**A**) This table shows the DCR in the three groups of antibiotic use duration, as follows: no prior antibiotics, <14 days, and ≥14 days. The DCR is compared between the <14 days and the ≥14 days groups of antibiotic use (*p* = 0.426) (**B**) The bar graph shows no significant difference in the DCR between the “No prior antibiotics” and the “Prior antibiotics” groups (33.3% vs. 23.4%, respectively). DCR, disease control rate; ICI, immune checkpoint inhibitor.

**Figure 2 cancers-15-05745-f002:**
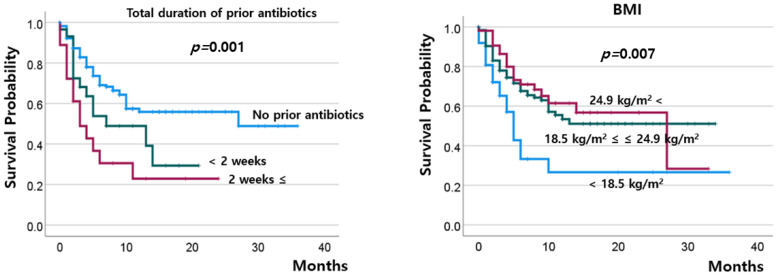
Overall survival according to the total duration of prior antibiotics and BMI.

**Table 1 cancers-15-05745-t001:** Baseline characteristics *n* = 215).

	All Patients	No Antibiotics	Prior Antibiotics	*p*
	*n* = 215	*n* = 168	*n* = 47	
Age, years	55 (28–86)	56 (28–86)	53 (30–79)	0.115
BMI, kg/m^2^	22.2 ± 3.9	22.4 ± 3.7	21.3 ± 4.5	0.084
ECOG				0.116
0 or 1	73 (34.0%)	62 (36.9%)	11 (23.4%)	
2 or 3	142 (66.0%)	106 (63.1%)	36 (76.6%)	
PDL1 expression				0.103
Positive	102 (47.4%)	75 (44.6%)	27 (57.4%)	
Negative	52 (24.2%)	46 (27.4%)	6 (12.8%)	
Unknown	61 (28.4%)	47 (28.0%)	14 (29.8%)	
dMMR/MSI-H				0.273
Positive	54 (25.1%)	46 (27.4%)	8 (17.1%)	
Negative	44 (20.5%)	35 (20.8%)	9 (19.1%)	
Unknown	117 (54.4%)	87 (51.8%)	30 (63.8%)	
Tumor sites *				0.082
Ovary	112 (52.1%)	90 (53.6%)	22 (46.8%)	
Cervix	53 (24.7%)	37 (22.0%)	16 (34.0%)	
Endometrium	36 (16.7%)	32 (19.0%)	4 (8.6%)	
Others	14 (6.5%)	9 (5.4%)	5 (10.6%)	
Prior lines of chemotherapy	3 (1–11)	3 (1–11)	2 (1–8)	0.310
Chemotherapy-free interval, m	2 (0–54)	2 (0–54)	2 (0–41)	0.966
Prior radiotherapy	110 (51.2%)	83 (49.4%)	27 (57.4%)	0.330
Type of ICI				0.998
Pembrolizumab	183 (85.1%)	143 (85.1%)	40 (85.1%)	
Nivolumab	32 (14.9%)	25 (14.9%)	7 (14.9%)	

The definition of PDL1 expression was based on PD-L1 immunohistochemistry by 22C3 pharmDx at a CPS ≥ 1 cutoff). The other tumor sites are the vagina and vulva. * Initial stage at diagnosis, according to the primary site described in Appendix A. dMMR, deficient mismatch repair; MSI-H, microsatellite instability-high; ICI, immune checkpoint inhibitors.

**Table 2 cancers-15-05745-t002:** Disease control rates (DCR) at 8–10 weeks based on BMI.

BMI	<18.5 kg/m^2^	≥18.5 kg/m^2^ and ≤24.9 kg/m^2^	>24.9 kg/m^2^	*p*
% (*n*)	17.2% (37)	58.1% (125)	24.7% (53)	0.065
DCR at 10 weeks	21.6% (8)	28.8% (36)	43.4% (23)	
Complete response	0	0.8% (1)	1.9% (1)	
Partial response	0	5.6% (7)	7.5% (4)	
Stable disease	21.6% (8)	22.4% (28)	34.0% (18)	
Progressive disease	78.4% (29)	71.2% (89)	56.6% (30)	

DCR, disease control rate; BMI, body mass index [calculated as weight (kg)/height (m)^2^]. *p* value < 0.05 was considered significant.

**Table 3 cancers-15-05745-t003:** Univariate and multivariate analyses of overall survival.

	Univariate Analysis			Multivariate Analysis		
	HR	95% CI	*p*	HR	95% CI	*p*
Prior antibiotics in 60 days						
No	1			1		
<14 days	1.803	1.021–3.182	0.042	1.516	0.848–2.708	0.160
≥14 days	2.817	1.543–5.141	0.001	2.286	1.210–4.318	0.011
BMI, kg/m^2^						
>24.9	1					
≥18.5 and ≤24.9	1.165	0.686–1.979	0.573	0.990	0.576–1.702	0.972
<18.5	2.381	1.265–4.481	0.007	1.573	0.808–3.065	0.183
Age						
<65 years	1			1		
≥65 years	0.738	0.409–1.330	0.312	0.874	0.469–1.629	0.673
MMR/MSI-H status						
Unknown	1			1		
pMMR/MSS	0.648	0.362–1.160	0.144	0.716	0.376–1.360	0.307
dMMR/MSI-H	0.429	0.239–0.768	0.004	0.632	0.331–1.206	0.164
Tumor sites						
Endometrium	1			1		
Non-endometrium	2.500	1.208–5.176	0.014	1.216	0.513–2.884	0.656
Chemotherapy-free interval						
≥6 months	1			1		
<6 months	2.282	1.240–4.202	0.008	2.007	1.055–3.819	0.034
ECOG						
1	1			1		
2 or 3	5.284	2.857–9.776	<0.001	4.677	2.497–8.762	<0.001

BMI, body mass index [calculated as weight (kg)/height (m)^2^; dMMR, deficient mismatch repair; pMMR, proficient mismatch repair; MSI-H, microsatellite instability high; MSS, microsatellite instability stable. Non-endometrium included the ovary, cervix, vagina, and vulva.

**Table 4 cancers-15-05745-t004:** Types and indications of the antibiotics used.

Reason for Antibiotic Use	Event	Types of Antibiotics	Duration of Use, Days (Median, Range)
UTI	12	Ciprofloxacin, Ceftriaxone, Tazoferan, Tigecycline, Fosfomycin, Cefditoren, Cefepime, Azithromycin, Sultamicillin	7 (2–60)
Postprocedure	8	Cefditoren, Tazoferan, Cefazolin, Ceftriaxone, Metronidazole, Tazoferan, Cefepime, Vancomycin, Cefotetan, Tazoferan	18 (8–68)
Peritonitis	7	Ceftriaxone, Metronidazole, Cefotaxime, Tazoferan, Prepenem, Cefotetan, Ertapenem, Vancomycin, Tigecycline, Levofloxacin	7 (3–61)
Septic shock	4	Ceftriaxone, Metronidazole, Meropenem, Vancomycin, Tazoferan, Cefotaxime, Ampicillin/sulbactam, Cefditoren	23 (13–48)
Colitis	4	Ceftriaxone, Cefuroxime, Ciprofloxacin, Moxifloxacin	10.5 (4–12)
Neutropenic fever	3	Cefepime, Tazoferan	11 (4–26)
Pneumonia	3	Tazoferan, Levofloxacin	6 (2–7)
Cellulitis	2	Amoxicillin, Ceftriaxone, Tazoferan, Meropenem, Levofloxacin, Vancomycin	26 (25–27)
Cholangitis	2	Cefotetan, Metronidazole, Moxifloxacin	17 (7–27)
Cholecystitis	1	Ciprofloxacin, Metronidazole	47
Complicated lymphocele	1	Tazoferna, Cefotetan	7
Unknown fever	1	Ciprofloxacin, Amoxicillin	5

UTI, urinary tract infection.

## Data Availability

Data available on request due to restrictions eg privacy or ethical. The data presented in this study are available on request from the corresponding author. The data are not publicly available due to institutional policy.

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
