# Peer review of "Adverse Effect of the Duration of Antibiotic Use Prior to Immune Checkpoint Inhibitors on the Overall Survival of Patients with Recurrent Gynecologic Malignancies"

_cancers, 2023, doi:10.3390/cancers15245745_

Round 1
Reviewer 1 Report (Previous Reviewer 2)
Comments and Suggestions for Authors
The present study investigated the effect of the duration of antibiotic use before checkpoint inhibitors (ICIs) therapy on oncological outcomes. The results showed that Prior use of antibiotics with a cumulative dose of more than 14 days was associated with reduced survival in recurrent gynecological malignancies. The study was overall well conducted. There are some points that should be further addressed by the authors.
1. The major concern is the significant heterogeneity of the included subjects. The effect of the ICIs therapy and the survival probability could be affected by different types of the malignancies and the different types of infection.
2. The results should be further validated in larger prospective cohorts.
Comments on the Quality of English LanguageModerate editing of English language required
Author Response
Thank you for your comments. It was helpful to revise the manuscript.
We made moderate revisions to the document in terms of language and grammar. Title was revised more clearly. In additin, redundant phrase was modified without change in contents.
Response to Reviewers' comments
- The major concern is the significant heterogeneity of the included subjects. The effect of the ICIs therapy and the survival probability could be affected by different types of malignancies and the different types of infection.
=>We discuss about the heterogeneity of study population as you mentioned. We made revision to describe more clearly. Heterogeneity of population may be a limitation of this study as we describe in Discussion in line #265-267. Heterogeneity is resulted from primary tumor site and types of infection not only but also age, BMI, sensitivity to prior therapy, type of prior therapy etc. These independent variables (primary tumor site not only but also age, BMI, sensitivity to prior therapy, type of prior therapy ) are included in statistical analysis by multivariate analysis as we decribed in Methods. We performed Cox proportional hazard model by multivariate analysis to evaluate multiple factors of the response to ICI therapy.
In line # 283-288, we described that various clinical situation of antibitics use. Heterogeneity exist as you pointed. Therefore, we described in the "Discussion"
- The results should be further validated in larger prospective cohorts.
I agree that the results need to be validated in larger prospective cohort as we dicussed in line #272-273.
Although limitations of this study, this study provide a small block of evidence to find prognostic factors of ICI therapy.

Reviewer 2 Report (Previous Reviewer 1)
Comments and Suggestions for Authors
The authors have correctly responded to the reviewer’s comments and properly amended manuscript. I have no further comment.
Author Response
Thank you for your previous comments. It was very helpful to improve our manuscript.
Round 2
Reviewer 1 Report (Previous Reviewer 2)
Comments and Suggestions for Authors
The authors have made corresponding revisions according to the reviewers' suggestions. I have no more questions.
Comments on the Quality of English LanguageMinor editing of English language required
This manuscript is a resubmission of an earlier submission. The following is a list of the peer review reports and author responses from that submission.
Round 1
Reviewer 1 Report
Comments and Suggestions for Authors
The authors retrospectively analyzed the correlation clinical factors (involving antibiotics usage) and therapeutic outcomes of ICIs in 215 patients with gynecological malignancies. Their results indicate that prior use of antibiotics for more than 14 days was associated with reduced survival in patients treated with ICIs. The major problem this study is that the tumor stages and histological subtypes are not considered in analyses of therapeutic outcomes. Obviously, tumor stages associate with prognosis and various types of tumors arises in gynecological organs and histological subtypes also associate with prognosis and effect of ICIs. In addition, association between ICIs and antibiotics is already reported as the authors described in Introduction. Therefore, the priority of this manuscript is not so high. The detail of assessment method for PD-L1 expression and MMR/MSI status should be documented. The Materials and Methods section should be documented with subheadings (such as Patients, Treatment, Assessment of PD-L1 expression, Assessment of MMR/MSI status, Assessment of ICI effect, and Statistical analyzes). ECOG is Eastern Cooperative Oncology Group Performance Status Scale? It should be clearly documented. In Supplementary Figure 7, “PLD1 Positive” should be “PDL1 positive”.
Reviewer 2 Report
Comments and Suggestions for Authors
The study aimed to evaluate the effect of the duration of antibiotic use before ICI therapy on oncological outcomes. The results showed that prior use of antibiotics with a cumulative dose of more than 14 days was associated with reduced survival in recurrent gynecological malignancies. There have been a lot of similar studies concerning the topic and the novelty and the clinical significance of the present findings were limited.
Comments on the Quality of English LanguageModerate editing of English language is needed.
Reviewer 3 Report
Comments and Suggestions for Authors
The authors aim to evaluate the effect of duration and timing of antibiotic use on ICI therapy in recurrent gynecological cancer and assess the effect of antibiotics on ICI response according to ORR and overall survival (OS) in women with recurrent gynecological cancers. Generally, this study is valuable and would generate wide readership in cancer research. However, there are couple of points which should be addressed.
1) Figure 1 consists of one figure and one table without any sub-labeling(like 1A and 1B), which easily gets confused. The description should be accurate. For example, "DCR at 8 to 10 weeks after ICIs was 33.3% of patients without prior antibiotics and 23.4% of patients with previous use of antibiotics (p=0.130) (Figures 1). " should change to "DCR at 8 to 10 weeks after ICIs was 33.3% of patients without prior antibiotics and 23.4% of patients with previous use of antibiotics (p=0.130) (Figure 1)."
2) All the figure legends are missing.